EMBO
Molecular Medicine

# Synergistic antibacterial effect of silver and ebselen against multidrug-resistant Gram-negative bacterial infections

Lili Zou[1], Jun Lu[1,2,*] (iD), Jun Wang[3], Xiaoyuan Ren[1], Lanlan Zhang[1], Yu Gao[4], Martin E Rottenberg[4] & Arne Holmgren[1,**] (iD)

## Abstract

Multidrug-resistant (MDR) Gram-negative bacteria account for a majority of fatal infections, and development of new antibiotic principles and drugs is therefore of outstanding importance. Here, we report that five most clinically difficult-to-treat MDR Gram-negative bacteria are highly sensitive to a synergistic combination of silver and ebselen. In contrast, silver has no synergistic toxicity with ebselen on mammalian cells. The silver and ebselen combination causes a rapid depletion of glutathione and inhibition of the thioredoxin system in bacteria. Silver ions were identified as strong inhibitors of *Escherichia coli* thioredoxin and thioredoxin reductase, which are required for ribonucleotide reductase and DNA synthesis and defense against oxidative stress. The bactericidal efficacy of silver and ebselen was further verified in the treatment of mild and acute MDR *E. coli* peritonitis in mice. These results demonstrate that thiol-dependent redox systems in bacteria can be targeted in the design of new antibacterial drugs. The silver and ebselen combination offers a proof of concept in targeting essential bacterial systems and might be developed for novel efficient treatments against MDR Gram-negative bacterial infections.

**Keywords** ebselen; multidrug-resistant Gram-negative bacteria; silver; synergistic antibacterial effect; thiol-dependent redox system
**Subject Categories** Microbiology, Virology & Host Pathogen Interaction; Pharmacology & Drug Discovery

## Introduction

The spread of multidrug-resistant (MDR) bacteria threatens modern medical treatment for infectious diseases (Walsh, 2003; Lewis, 2013). This is particularly true for MDR Gram-negative bacteria, which account for up to 80% of severe bacterial infections in the clinic, with very few antibiotics that can currently combat them or that are being developed (Walsh, 2003; Lewis, 2013). Current antibiotics are mainly developed through the mechanisms of inhibition of bacterial cell wall, membrane, DNA, and protein synthesis (Walsh, 2003). Here we have explored a new antibiotic strategy based on inhibition of bacterial thiol-dependent redox system.

There are two major thiol-dependent enzyme systems in prokaryotic and eukaryotic cells based on thioredoxin (Trx) and glutathione (GSH), which transfer electrons from NADPH to their substrates via thioredoxin reductases (TrxR) and glutathione reductases (GR) (Holmgren, 1989; Ritz & Beckwith, 2001; Lillig & Holmgren, 2007). These two systems are critical for DNA synthesis, defense against oxidative stress, repair of oxidized proteins, and post-translational modifications such as protein *S*-glutathionylation, or *S*-nitrosylation, which are important for many cellular processes (Holmgren, 1989; Ritz & Beckwith, 2001; Lillig & Holmgren, 2007).

Bacterial thiol-dependent redox systems have notable differences in components, enzyme structures, and reaction mechanisms compared with corresponding systems in the mammalian host (Holmgren, 1989; Nozawa *et al*, 1989; Ritz & Beckwith, 2001; Lillig & Holmgren, 2007; Lu *et al*, 2013a; Lu & Holmgren, 2014). For example, bacterial TrxR is an enzyme with a cysteine-containing active site, while highly efficient mammalian TrxR is structurally different with a selenocysteine (Sec)-containing active site. In addition to this property, the combination of highly antioxidant efficiency from glutathione peroxidases (GPx), peroxiredoxins (Prxs), and catalases (CAT) confer mammalian hosts an absolute advantage to maintain the reactive oxygen species (ROS) at a low level in comparison with bacteria (Morones-Ramirez *et al*, 2013a; Lu & Holmgren, 2014; Harbut *et al*, 2015). Thus, all these differences guarantee bacterial redox systems are appropriate potential targets by specific antibiotics.

Trx system is ubiquitous in bacteria, whereas GSH systems are lacking in some specific bacteria (Lu & Holmgren, 2014). Nearly all

1 Division of Biochemistry, Department of Medical Biochemistry and Biophysics, Karolinska Institutet, Stockholm, Sweden
2 School of Pharmaceutical Sciences, Southwest University, Chongqing, China
3 Translational Neuroscience & Neural Regeneration and Repair Institute/Institute of Cell Therapy, The First Hospital of Yichang, Three Gorges University, Yichang, China
4 Department of Microbiology, Tumour and Cell Biology, Karolinska Institutet, Stockholm, Sweden
*Corresponding author. Tel: +86 13594206128; E-mails: jun.lu@ki.se; junlu@swu.edu.cn
**Corresponding author. Tel: +46 70 6467686; Fax: +46 8 7284716; E-mail: arne.holmgren@ki.se

Gram-negative bacteria contain GSH and glutaredoxin (Grx) (GSH-positive bacteria), whereas most Gram-positive bacteria lack GSH (GSH-negative bacteria) (Martin, 1995; Wilkinson *et al*, 2011; Lu & Holmgren, 2014). We previously reported that ebselen [2-phenyl-1,2 benzisoselenazol-3(2H)-one], a substrate of mammalian thioredoxin reductase (TrxR) but a competitive inhibitor of bacterial TrxR, displays selectively antibacterial activity toward GSH-negative bacteria (Lu *et al*, 2013a), with no inhibitory effect on GSH-positive bacteria, for example, *Escherichia coli* (Nozawa *et al*, 1989).

Here we present a new antibiotic strategy selectively targeting bacterial thiol-dependent redox systems via strong bactericidal effect of silver and ebselen in synergistic combination against GSH-positive bacterial infections, particularly on MDR Gram-negative bacteria. It has been reported that the antibacterial activity of silver might be associated with disruption of disulfide bond formation and enhanced production of ROS (Russell & Hugo, 1994; Kohanski *et al*, 2007), but the exact mechanisms are still unclear. In this paper, we elucidated that silver ions were strong inhibitors of both *E. coli* Trx and TrxR, and the combination with ebselen depleted GSH and gave a steep rise in ROS generation. Furthermore, we found that the presence of ebselen dramatically decreased the antibacterial concentration of silver, with highly significant selective toxicity on bacteria over mammalian cells. This selective toxicity should facilitate the systemic medical application of silver in the treatment of MDR Gram-negative bacteria.

## Results

### Combination of silver with ebselen exhibited selective synergistic toxicity against bacteria

The effect of silver nitrate with ebselen in combination on the growth of Gram-negative model bacteria, *E. coli*, was investigated in the microplates. DHB4 overnight cultures were diluted 1:1,000 times in Luria-Bertani (LB) medium and treated with ionic silver ($Ag^+$) as a nitrate salt ($AgNO_3$) for 16 h. $Ag^+$ alone inhibited *E. coli* growth with a minimal inhibition concentration (MIC) of 42 μM after 16-h treatment, while the addition of 2 μM ebselen dramatically decreased the MIC of $Ag^+$ to 4.2 μM ($P = 0.000028$) (Fig 1A). Meanwhile, 5 μM $Ag^+$ and 2.5 μM ebselen in combination showed no synergistic toxicity on human HeLa cells ($P = 0.98$) (Fig 1B). In addition, the toxicity of ebselen itself (2, 4, 8 μM) on bacterial and mammalian cells was similar (Fig 1A and B) with no effects on bacterial growth (Fig EV1). These results indicate that treatment of $Ag^+$ with ebselen in combination exhibits significant selective synergistic toxicity on bacteria over mammalian cells, and the dramatic decrease in MIC of silver against bacteria in the presence of ebselen make the systemic medical use of silver feasible.

The large-scale growth inhibition of *E. coli* by $Ag^+$ with ebselen in combination was also observed in shaking testing 15-ml tubes. *Escherichia coli* DHB4 cells were grown until an $OD_{600\ nm}$ of 0.4, and treated with 5 μM $Ag^+$ and serial concentrations of ebselen (0, 20, 40, 80 μM). The growth curves showed a synergistic bacteriostatic effect of $Ag^+$ with ebselen in combination in LB medium (Fig 2A), and the synergistic bactericidal effect of 5 μM $Ag^+$ and 80 μM ebselen in combination was further confirmed by the colony formation assay on LB-agar plates (Fig 2B). Meanwhile, only 80 μM

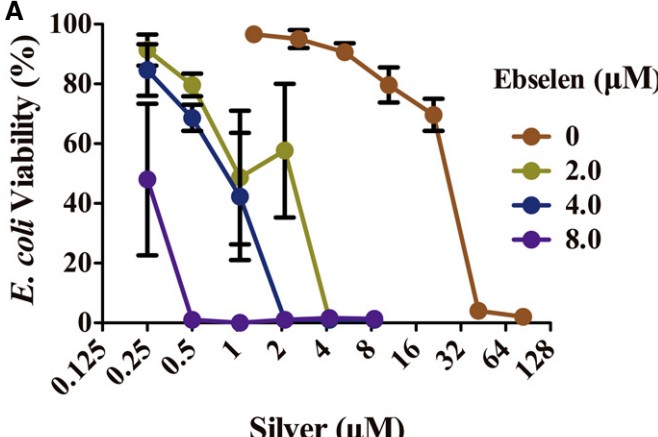

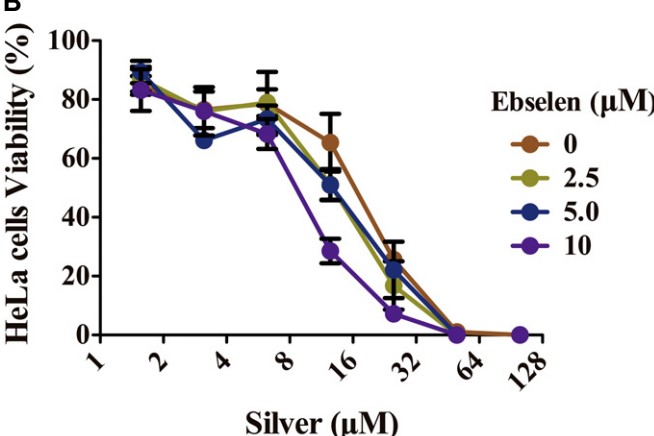

**Figure 1.  Effects of silver with ebselen in combination on the growth of *Escherichia coli* and HeLa cells.**

A  Synergistic effect of ebselen with silver nitrate ($AgNO_3$) in combination on the growth of *E. coli*. *Escherichia coli* DHB4 overnight cultures were diluted 1:1,000 into 100 μl of LB medium in 96 micro-well plates, and treated with 100 μl serial dilutions of ebselen and $AgNO_3$ in combination for 16 h, and cell viability was determined by measuring $OD_{600\ nm}$. $Ag^+$ alone inhibited *E. coli* growth with a minimal inhibition concentration (MIC) of 42 μM after 16-h treatment, while 2 μM ebselen dramatically decreased the MIC of $Ag^+$ to 4.2 μM ($P = 0.000028$, Student's *t*-test).

B  Effects of ebselen with $AgNO_3$ in combination on the growth of HeLa cells. HeLa cells were treated with serial concentrations of ebselen and $AgNO_3$ for 24 h, and cell toxicity was detected by MTT assay. 5 μM $Ag^+$ and 2.5 μM ebselen in combination showed no synergistic toxicity on human HeLa cells ($P = 0.98$, Student's *t*-test).

Data information: Data are presented as means ± SD of three independent experiments.

ebselen itself could inhibit *E. coli* growth in first 8 h, and gains back into normal 12 h post-treatment (Fig EV2). While 40 μM ebselen or 5 μM $Ag^+$ alone did not inhibit bacterial growth, $Ag^+$ with ebselen in combination resulted in strong inhibition of *E. coli* growth (Fig 2A and B). In line with this, 5 μM $Ag^+$ and 20 μM ebselen in combination enhanced the frequency of propidium iodide (PI) staining ($P = 0.00083$) (Fig 2C and D). PI is a membrane-impermeable fluorescent dye that has been widely used to detect permeation of cell membrane and dead cells. Above all, these results indicate that

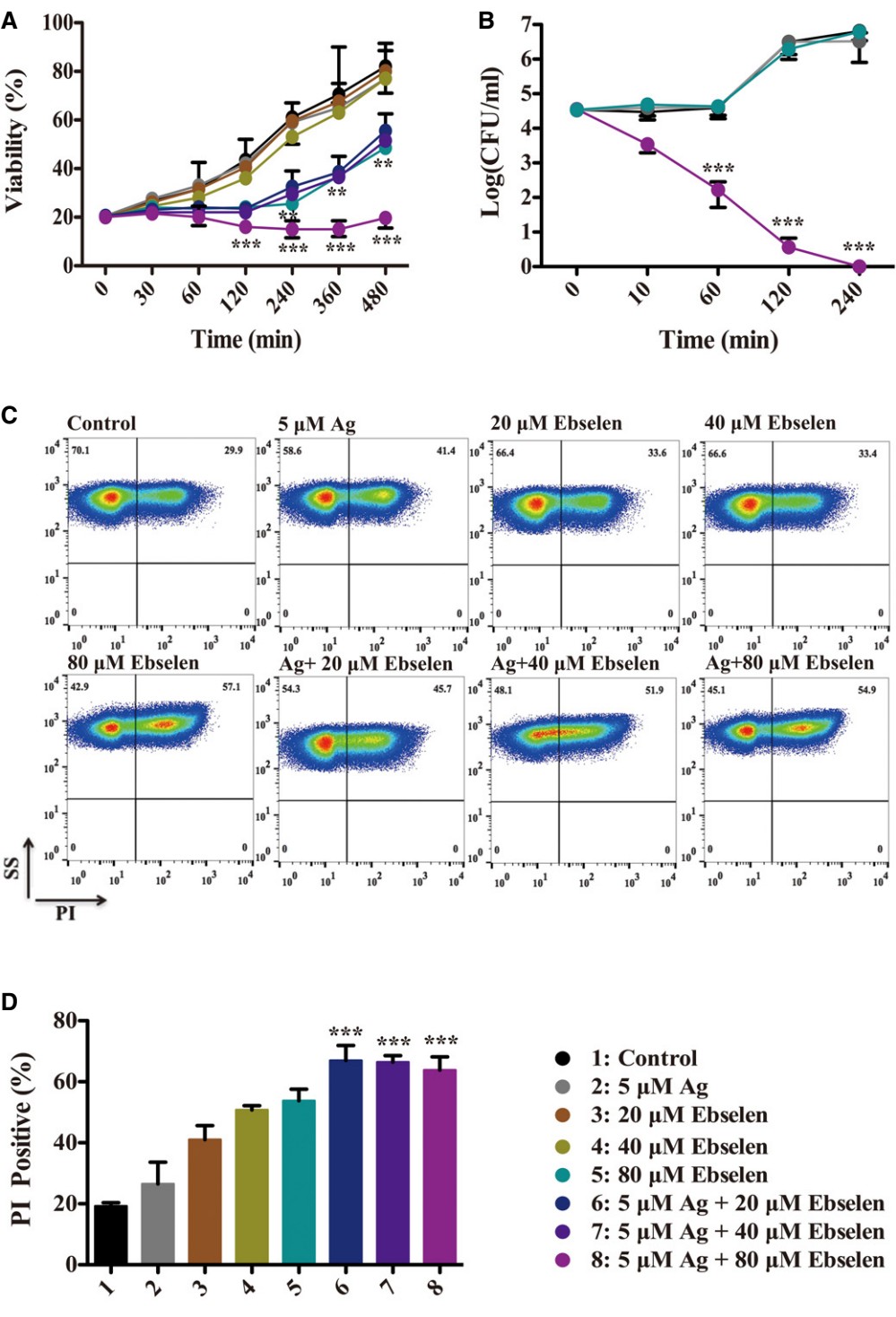

**Figure 2. Silver with ebselen in combination exhibited synergistic bactericidal effect.**

*Escherichia coli* DHB4 grown to $OD_{600\ nm}$ of 0.4 were treated with serial dilutions of ebselen and $AgNO_3$ in combination.

A Cell viability was represented by measuring $OD_{600\ nm}$. The growth curves showed a synergistic bacteriostatic effect of $Ag^+$ with ebselen in combination in LB medium. 5 µM $Ag^+$ and 40 µM ebselen in combination inhibited *E. coli* growth 480 min post-treatment (**$P = 0.0075$).

B Changes of colony forming units of *E. coli* DHB4 on LB plates 0, 10, 60, 120, and 240 min post-treatment. The synergistic bactericidal effect of 5 µM $Ag^+$ and 80 µM ebselen in combination was confirmed by the colony formation assay on LB-agar plates. 5 µM $Ag^+$ and 80 µM ebselen in combination killed the majority of *E. coli* 60 min post-treatment (***$P = 0.00021$).

C, D FACS plots (C) and mean ± SD (D) of propidium iodide (PI)-stained *E. coli* DHB4. 5 µM $Ag^+$ and 20 µM ebselen in combination enhanced the frequency of propidium iodide (PI) staining (***$P = 0.00083$).

Data information: Data are presented as means ± SD of three independent experiments. **$P < 0.01$, ***$P < 0.001$ (Student's *t*-test).

Ag$^+$ and ebselen in combination exhibited a selective synergistic effect on bacteria.

## Clinically isolated five most difficult-to-treat MDR Gram-negative pathogens were sensitive to Ag$^+$ with ebselen in combination

There are five most difficult-to-treat MDR Gram-negative pathogen species in the clinic, which are also typical GSH-positive bacteria: *Klebsiella pneumonia*, *Acinetobacter baumannii*, *Pseudomonas aeruginosa*, *Enterobacter cloacae*, and *Escherichia coli*. Two strains of each species were isolated, and overnight cultures were diluted 1:1,000 times in LB medium, and treated with Ag$^+$ with a serial concentration of ebselen in combination for 16 h. The synergistic bactericidal effects of Ag$^+$ with ebselen in combination against all 10 tested strains were observed (Table 1). Among these five species, *A. baumannii* and *E. cloacae* are very readily formed drug-resistant strains, which are needed to be treated by carbapenems (our current "last good line" antibiotics) or the fourth-generation cephalosporin in the clinic, including imipenem, cefepime, and cefotaxime. The isolated imipenem, cefepime, and cefotaxime-resistant *A. baumannii* (AB-1/2) and *E. cloacae* (ECL-1) strains were identified (Appendix Tables S1 and S2) and were sensitive to Ag$^+$ with ebselen in combination (Table 1). These results indicate that Ag$^+$ with ebselen in combination might be "the last life-saving straw" that are active against a range of bacteria with existing resistance, which would increase the correct chance for empirically prescribed therapy, even for infections resistant to our current antibiotics.

## Ag$^+$ with ebselen in combination directly disrupted bacterial Trx and GSH systems

Since Ag$^+$ and ebselen are generally thought to be thiol-targeting agents, we detected bacterial TrxR or Trx activities in cells treated by Ag$^+$ with ebselen in combination (Fig 3A and B). While TrxR and Trx activities in cell extracts were not influenced by either Ag$^+$ or ebselen alone, 5 μM Ag$^+$ and 20 μM ebselen in combination resulted in a dramatic loss of TrxR ($P = 0.00018$) and Trx ($P = 0.0036$) activities (Fig 3A and B). Consistent with this observation, the redox state of Trx1 measured by redox Western blot was also affected by treatment of Ag$^+$ with ebselen in combination. Trx1 was mostly in reduced form in untreated bacteria, which became oxidized upon treatment of drugs in combination (Fig 3C). We also used a Trx2 antibody to detect oxidized Trx2 in the experiment to

investigate the effect of treatment on the redox state of Trxs. Reduced Trx2 could not be detected by this antibody probably because of the blockage of the recognition site. None of the oxidized Trx2 was observed upon the treatment, while the positive control diamide-oxidized Trx2 was detected (Fig 3D). These results showed that Trx2 was less sensitive to the treatment compared to Trx1. In addition, the protein levels of Trx1 and Trx2 were not affected by the 10-min treatment with Ag$^+$ and ebselen combination (Fig 3C and D).

The addition of 5 μM Ag$^+$ and 20, 40, or 80 μM ebselen also decreased the GSH levels. 5 μM Ag$^+$ and 20 μM ebselen in combination treatment depleted the functional GSH in 10 min compared with control ($P = 0.000021$) (Fig 3E). Ebselen alone at 80 and 40 μM also reduced GSH levels, albeit less efficiently than the corresponding drugs in combination at the same concentration ($P = 0.000076$ and $0.000029$). Instead, neither 5 μM Ag$^+$ nor 20 μM ebselen modulated GSH levels in the conditions tested compared with control ($P = 0.081$ and $0.712$) (Fig 3E).

Whether Ag$^+$ with ebselen in combination decreased or depleted GSH could affect protein *S*-glutathionylation was further explored (Fig 3F). Protein *S*-glutathionylation was decreased in Ag$^+$ with ebselen-treated bacteria, but not in those incubated only with 5 μM Ag$^+$ or 20 μM ebselen alone. Thus, the presence of 5 μM Ag$^+$ decreased protein *S*-glutathionylation when combined with 20 μM ebselen reflecting the loss of GSH (Fig 3F).

Since Trx and GSH/Grx are major thiol-dependent systems, we investigated the effects of Ag$^+$ with ebselen in combination on Trx or GSH system-deficient *E. coli* redox mutants. *Escherichia coli* mutants lacking GSH system components ($gshA^-$) and living on Trx and TrxR were more sensitive to Ag$^+$ and ebselen treatment compared with the wild type (WT) (Tables 2 and 3, and Appendix Table S3). All results showed that Ag$^+$ with ebselen in combination has strong synergistic effects on bacterial Trx and GSH systems, and targeting thiol-dependent systems as a novel antibiotic strategy.

## Silver irreversibly inhibits bacterial Trx and TrxR activities

Cellular TrxR and Trx1 enzyme activities were decreased, while the corresponding protein levels were unaltered by the treatment of Ag$^+$ with ebselen in combination, suggesting that TrxR and Trx were inhibited. Since ebselen is a known reversible competitive inhibitor of bacterial TrxR, we investigated the effect of Ag$^+$ on the activity of *E. coli* TrxR and Trx. When 100 nM of NADPH-pre-incubated *E. coli*

**Table 1.   MIC of silver (μM) in the presence of ebselen against different multidrug-resistant Gram-negative species.**

| Ebselen (μM) | MIC of silver (μM) in the presence of ebselen against multidrug-resistant Gram-negative species | | | | | | | | | | Others | |
| --- | --- | --- | --- | --- | --- | --- | --- | --- | --- | --- | --- | --- |
| | KP-1 | KP-2 | AB-1 | AB-2 | PA-1 | PA-2 | ECL-1 | ECL-2 | ECO-1 | ECO-2 | ECO-3 | ECO-4 |
| 0 | 80 | 80 | 80 | 80 | 80 | 80 | 80 | 80 | 40 | 80 | 40 | 40 |
| 1 | 80 | 40 | 80 | 80 | 80 | 80 | 40 | 40 | 20 | 80 | 40 | 20 |
| 2 | 40 | 20 | 40 | 40 | 20 | 40 | 20 | 40 | 10 | 40 | 20 | 10 |
| 4 | 10 | 20 | 10 | 20 | 20 | 20 | 20 | 10 | 5 | 10 | 10 | 5 |

KP-1: *Klebsiella pneumoniae* (*K. pneumoniae*) subsp. *pneumoniae* 13#; KP-2: *K. pneumoniae* subsp. *pneumoniae* 0322#; AB-1: *Acinetobacter baumannii* (*A. baumannii*) H#; AB-2: *A. baumannii* 0361#; PA-1: *Pseudomonas aeruginosa* (*P. aeruginosa*) 1298#; PA-2: *P. aeruginosa* 0009#; ECL-1: *Enterobacter cloacae* (*E. cloacae*) 0431#; ECL-2: *E. cloacae* 2301#; ECO-1: *Escherichia coli* (*E. coli*) 1139#; ECO-2: *E. coli* 2219#; ECO-3: *E. coli* ZY-1; ECO-4: MG1655 (ATCC 700926).

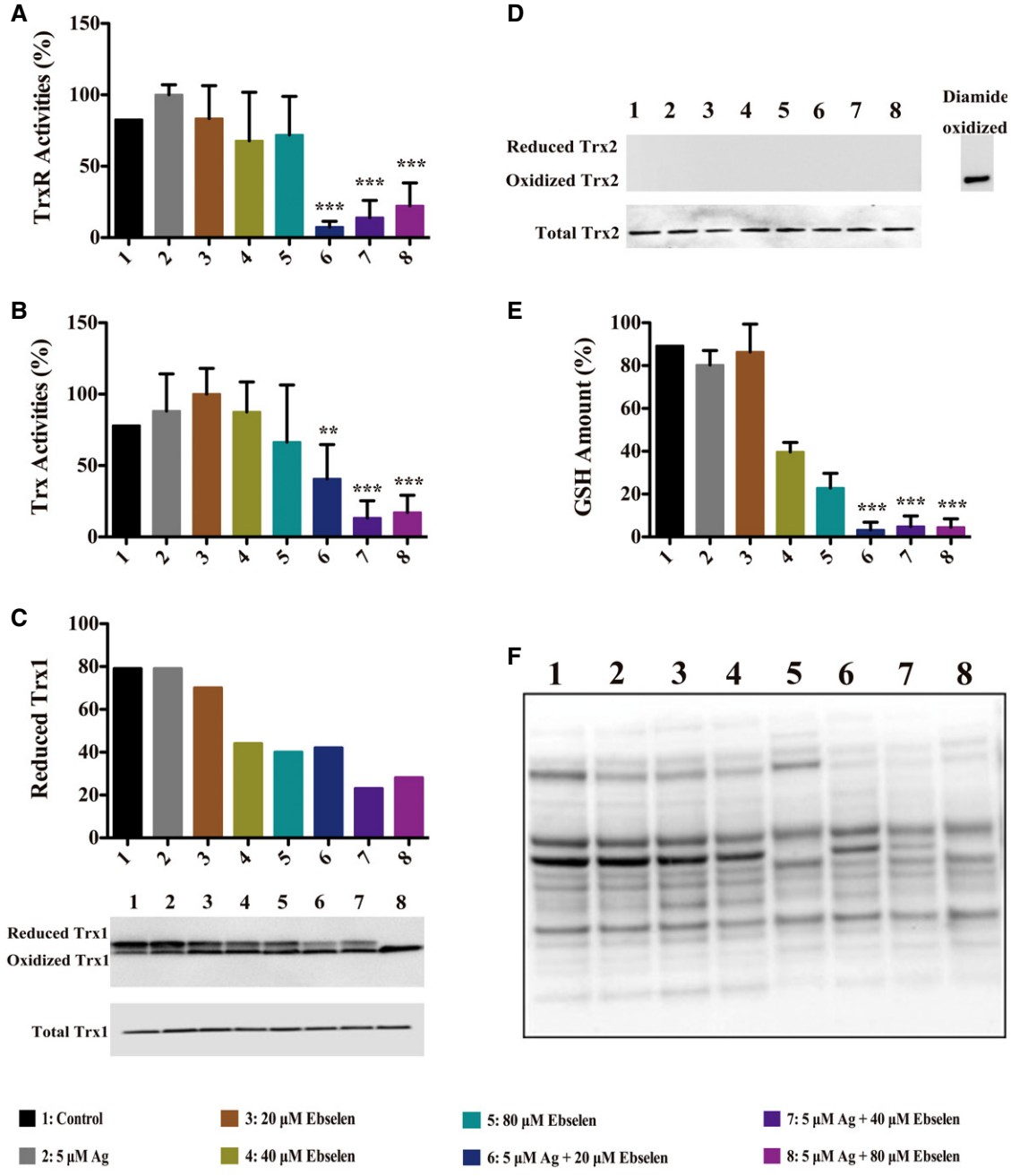

**Figure 3. Silver with ebselen in combination directly disrupted bacterial Trx and GSH systems.**

*Escherichia coli* DHB4 grown to OD$_{600\ nm}$ of 0.4 were treated with serial dilutions of ebselen and AgNO$_3$ in combination.

A   TrxR activities were assayed using DTNB reduction in the presence of Trx in *E. coli* extracts, 50 mM Tris–HCl (pH 7.5), 200 μM NADPH, 1 mM EDTA, 1 mM DTNB, in the presence of 100 nM *E. coli* TrxR. 5 μM Ag$^+$ and 20 μM ebselen in combination resulted in a dramatic loss of TrxR activities (***$P$ = 0.00018).

B   Trx activities were assayed using DTNB reduction in the presence of Trx in *E. coli* extracts, 50 mM Tris–HCl (pH 7.5), 200 μM NADPH, 1 mM EDTA, 1 mM DTNB, 5 μM *E. coli* Trx. 5 μM Ag$^+$, and 20 μM ebselen in combination resulted in a dramatic loss of Trx activities (**$P$ = 0.0036).

C   Changes of Trx1 redox state in *E. coli* upon ebselen and AgNO$_3$ treatment. *Escherichia coli* were precipitated in 5% TCA and alkylated with 15 mM AMS, and the percent of reduced Trx1 were analyzed by Western blot.

D   Changes of Trx2 redox state in *E. coli* upon ebselen and AgNO$_3$ treatment. *Escherichia coli* were precipitated in 5% TCA and alkylated with 15 mM AMS, diamide-oxidized Trx2 was used as a Trx2 positive control, and the percent of reduced Trx2 were analyzed by Western blot.

E   GSH amounts were measured by GR-coupled DTNB reduction assay in *E. coli* extracts, 50 mM Tris–HCl (pH 7.5), 200 μM NADPH, 1 mM EDTA, 1 mM DTNB, 50 nM GR. 5 μM Ag$^+$, and 20 μM ebselen in combination depleted the functional GSH in 10 min compared with control (***$P$ = 0.000021).

F   Changes of proteins S-glutathionylation in *E. coli*. Cells were cultured, washed, and re-suspended in lysis buffer containing 30 mM IAM. After lysed by sonication, Western blotting assay was performed with IgG2a mouse monoclonal antibody (VIROGEN, 101-A/D8) for glutathione–protein complexes.

Data information: Data are presented as means ± SD of three independent experiments. **$P$ < 0.01, ***$P$ < 0.001 (Student's *t*-test).

**Table 2. MIC of silver (µM) in the presence of ebselen against *Escherichia coli* DHB4 mutants.**

| Ebselen (µM) | MIC of silver (µM) in the presence of ebselen against *Escherichia coli* DHB4 redox mutants | | | | | | | | | | |
|---|---|---|---|---|---|---|---|---|---|---|---|
| | WT | *trxA⁻* | *trxB⁻* | *trxC⁻* | *trxA⁻B⁻C⁻* | *oxyR⁻* | *gshA⁻* | *trxA⁻gshA⁻* | *gor⁻* | *gor⁻grxA⁻B⁻C⁻* | *grxA⁻trxA⁻* |
| 0 | 40 | 40 | 40 | 40 | 40 | 20 | 20 | 20 | 40 | 20 | 20 |
| 1 | 10 | 10 | 10 | 10 | 10 | 5 | 5 | 5 | 10 | 10 | 10 |
| 2 | 5 | 2.5 | 2.5 | 5 | 2.5 | 1.25 | 2.5 | 2.5 | 5 | 2.5 | 2.5 |
| 4 | 1.25 | 1.25 | 0.625 | 1.25 | 1.25 | 0.625 | 0.625 | 0.625 | 1.25 | 0.625 | 0.625 |

**Table 3. MIC of ebselen (µM) in the presence of silver against *Escherichia coli* DHB4 mutants.**

| Silver (µM) | MIC of ebselen (µM) in the presence of ebselen against *Escherichia coli* DHB4 redox mutants | | | | | | | | | | |
|---|---|---|---|---|---|---|---|---|---|---|---|
| | WT | *trxA⁻* | *trxB⁻* | *trxC⁻* | *trxA⁻B⁻C⁻* | *oxyR⁻* | *gshA⁻* | *trxA⁻gshA⁻* | *gor⁻* | *gor⁻grxA⁻B⁻C⁻* | *grxA⁻trxA⁻* |
| 0 | 80 | 80 | 80 | 80 | 80 | 40 | 40 | 40 | 80 | 40 | 40 |
| 0.625 | 8 | 8 | 8 | 8 | 8 | 4 | 4 | 4 | 8 | 8 | 8 |
| 1.25 | 4 | 4 | 2 | 4 | 4 | 2 | 2 | 2 | 4 | 2 | 2 |
| 2.5 | 4 | 2 | 2 | 4 | 2 | 2 | 2 | 2 | 2 | 2 | 2 |
| 5 | 2 | 1 | 1 | 2 | 1 | 1 | 1 | 1 | 1 | 1 | 1 |
| 10 | 1 | 1 | 0.5 | 1 | 0.5 | 0.5 | 0.5 | 0.5 | 1 | 0.5 | 0.5 |
| 20 | 0.5 | 0.5 | 0.5 | 0.5 | 0.5 | 0 | 0 | 0 | 0.5 | 0 | 0 |
| 40 | 0 | 0 | 0 | 0 | 0 | 0 | 0 | 0 | 0 | 0 | 0 |

TrxR was incubated with $Ag^+$, the $IC_{50}$ was about 50 nM (Fig 4A), similar to the gold compound auranofin (Harbut *et al*, 2015). To detect whether *E. coli* TrxR can be specifically inhibited by $Ag^+$, the enzyme was incubated with $Ag^+$ in the presence of reduced *E. coli* Trx1, the inhibitory efficiency toward TrxR decreased (Fig 4A). This indicated that Trx also reacted with $Ag^+$ and played a protective role for the TrxR. Using fluorescence spectroscopy, we further verified that $Ag^+$ interacted with reduced Trx1 and changed its fluorescence spectra (Fig 4B). Incubation with 1–10 µM $Ag^+$ increased the tryptophan fluorescence intensity of 10 µM Trx1. Meanwhile, the fluorescent intensity of 10 µM Trx1 decreased when treated with 20–100 µM $Ag^+$ (Fig 4B). In line with this, the activity of Trx decreased along with the increase in $Ag^+$ concentration (Fig 4C). The inhibition of Trx1 by $Ag^+$ was irreversible since the Trx1 activity was not recovered after desalting ($P = 0.00021$) (Fig 4D). This indicated that $Ag^+$ formed a tight complex with the sulfhydryl groups in reduced *E. coli* Trx1. All these results show that silver irreversibly inhibits bacterial Trx and TrxR activities.

### ROS is a determining factor for synergistic bactericidal effect of $Ag^+$ and ebselen

One major function of GSH and Trx systems is to scavenge ROS to keep cellular redox balance and protect against oxidative stress. The inhibition of the Trx system and depletion of GSH may responsible for the elevation of ROS. To determine whether increased ROS levels accounted for the bactericidal effect, we therefore determined ROS levels in $Ag^+$ and ebselen-treated cells. Treatment with either 5 µM $Ag^+$ or 20 µM ebselen alone did not change ROS concentrations, while the combination of 5 µM $Ag^+$

and 20 µM ebselen resulted in increased levels of ROS ($P = 0.00012$) (Fig 5A and B). Further, the enhancement of $H_2O_2$ levels caused by the treatment with 5 µM $Ag^+$ and 20 µM ebselen in combination was also verified by Amplex Red method (Dwyer *et al*, 2014) ($P = 0.00057$) (Fig 5C). In addition, *E. coli* mutants lacking OxyR components (*OxyR⁻*) that impair *E. coli* dehydratase clusters from $H_2O_2$ injury were more sensitive to $Ag^+$ and ebselen treatment in combination compared with the wild type (WT) (Tables 2 and 3, and Appendix Table S3). All results showed that lethality of $Ag^+$ with ebselen against bacteria is accompanied by ROS generation.

### Silver with ebselen in combination potentiated bactericidal antibiotics *in vivo*

The bactericidal effect of $Ag^+$ with ebselen in combination was also observed in LB medium containing heparinized mice blood (Fig EV3). To investigate whether the bactericidal activity of $Ag^+$ with ebselen in combination is also efficient *in vivo*, mice were infected i.p. with $6.0 \times 10^7$ or $1.7 \times 10^6$ MDR *E. coli* ZY-1 (Appendix Table S4), modeling an acute and a mild peritonitis, respectively. The $LD_{50}$ of *E. coli* ZY-1 administered i.p. was $1.3 \times 10^7$ CFU/ml. One hour (acute model) or 24 h (mild model) after infection, mice were treated i.p. with ebselen, $Ag^+$, or the drugs in combination, or remained untreated. The combination of $Ag^+$ and ebselen led to a significant reduction in bacterial load compared with the control in the mild peritonitis model. Mice treated with ebselen alone or left untreated showed similar levels of bacteria load after 36 h of infection with $6.0 \times 10^7$ *E. coli*, whereas treatment with $Ag^+$ with ebselen in combination achieved a

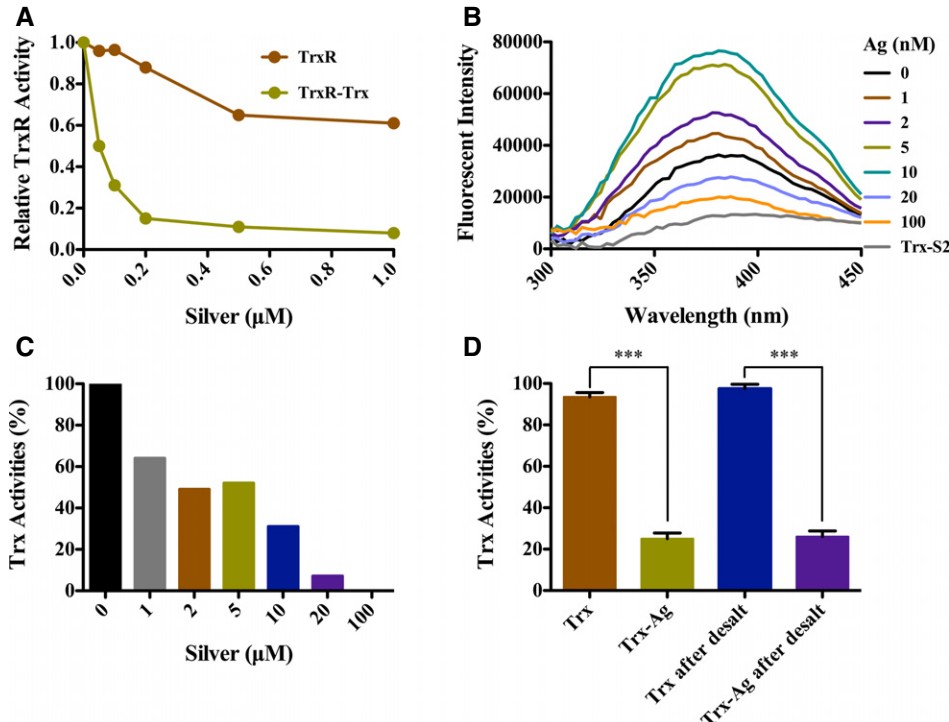

**Figure 4. Inhibitory effects of silver on *E. coli* Trx system *in vitro*.**

A   Inhibition of *E. coli* TrxR by $AgNO_3$. Pure recombinant 100 nM TrxR and 5 μM Trx mixture were incubated with $AgNO_3$ solution in the presence of 200 μM NADPH, and then, their activities were detected by DTNB reduction assay.

B   Fluorescence spectra of a complex between reduced *E. coli* 10 μM Trx1 with $AgNO_3$. Reduced 10 μM *E. coli* Trx1 protein was incubated with a serial concentration of $AgNO_3$ solution, and the fluorescent spectra was detected with an excitation wavelength at 280 nm. Oxidized Trx1 (Trx-$S_2$) was used as a control.

C   Inhibition of Trx by $AgNO_3$. After the treatment described in (B), Trx activity was assayed by a DTNB method in the presence of *E. coli* Trx1.

D   Inhibition reversibility of *E. coli* Trx1 by $AgNO_3$. Silver-inhibited *E. coli* Trx1 was passed through a desalting column to remove small molecules, and then, Trx activity was measured. *Escherichia coli* Trx1 without the inhibition was used as a control. The inhibition of Trx1 by $Ag^+$ was irreversible since the Trx1 activity was not recovered after desalting (***$P = 0.00021$).

Data information: In (D), data are presented as means $\pm$ SD of three independent experiments. ***$P < 0.001$ (Student's $t$-test).

100-fold reduction compared with control ($P = 0.0055$) (Fig 6A). Additionally, 80% of mice treated with $Ag^+$ with ebselen in combination survived in the acute peritonitis mice model, compared with 30% in control group (Fig 6B). These findings demonstrated the effective antibacterial effect of $Ag^+$ and ebselen against MDR Gram-negative pathogen *in vivo*.

$Ag^+$ or ebselen alone has been proven to be safe in previous studies (Kohanski *et al*, 2007; Wilkinson *et al*, 2011). To test the toxicity of the combination, we divided five mice per group, which were treated with 6 mg $AgNO_3$/kg body weight in combination with serial concentrations of ebselen (10, 15, 20, and 25 mg ebselen/kg body weight). Mice were observed for 7 days and remained viable with no mortality. Then, we determined whether the doses of $Ag^+$ and ebselen used in this study had a toxic effect on the mammalian host. We evaluated the effect of 25 mg ebselen/kg and 6 mg $AgNO_3$/kg body weight in combination on mice, by measuring key metabolite and enzyme concentrations using a Blood Chemistry Analyzer after treatment for 6, 24, and 48 h. The density of lymphocytes and monocytes and some enzymes, such as alanine transaminase in mice treated with $Ag^+$ and ebselen in combination, were reduced at the initial point (6 h); however, their values gain back

into normal at 24 h post-treatment (Table 4), indicating that there is a stress response upon the initial treatment. These results demonstrated that $Ag^+$ and ebselen were not toxic for mice at the conditions tested.

## Discussion

Multiple factors drive the development and spread of MDR bacteria. WHO showed that if we fail to act on this issue then an additional 10 million lives would be lost each year by 2050, at a cost to the world economy of 100 trillion USD. A concerted focus since the 1990s on tackling rising MDR Gram-positive bacteria within US and European healthcare systems appears to have been instrumental in stimulating the relatively large numbers of products targeting Gram-positive bacteria in recent years. The emergence of MDR Gram-negative bacteria presents a great threat to human life and is a challenge for modern medicine. The five most difficult-to-treat Gram-negative bacteria, *K. pneumonia*, *A. baumannii*, *P. aeruginosa*, *E. cloacae*, and *E. coli*, can be deadly in the clinic, causing urinary tract infections, life-threatening pneumonia and septicemia, with

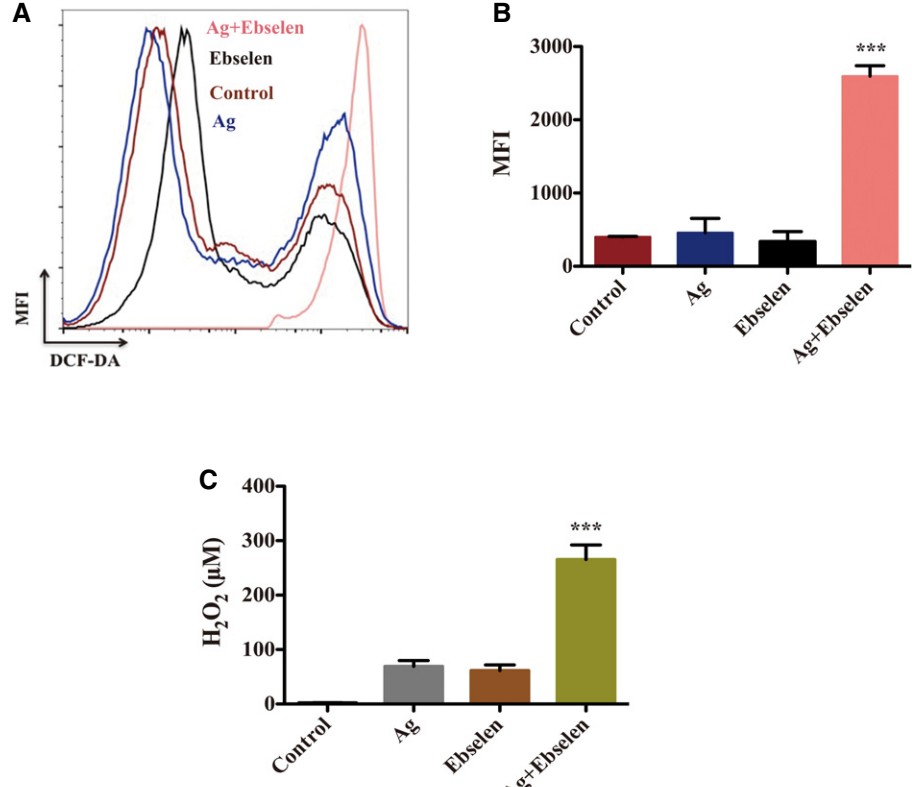

**Figure 5.  ROS was a determining factor for synergistic bactericidal effect of silver and ebselen.**

A, B    *E. coli* DHB4 grown to $OD_{600\ nm}$ of 0.4 were treated with 20 μM ebselen and 5 μM $AgNO_3$, and FACS histograms (A) and mean MFI ± SD (B) of $H_2$DCF-DA-stained *E. coli* were detected. ROS level was detected by flow cytometry (CyAn adp, Beckman Coulter). Treatment with either 5 μM $Ag^+$ or 20 μM ebselen alone did not change ROS concentrations, while the combination of 5 μM $Ag^+$ and 20 μM ebselen resulted in increased levels of ROS (***$P$ = 0.00012; Student's *t*-test).

C    Detection of $H_2O_2$ using the Amplex® Red Hydrogen Peroxide/Peroxidase Assay Kit (Invitrogen). Reactions containing 50 μM Amplex® Red reagent, 0.1 U/ml HRP, and the samples in 50 mM sodium phosphate buffer, pH 7.4, were incubated for 30 min at room temperature and detected with absorbance at 560 nm. Background determined for a non-$H_2O_2$ control reaction has been subtracted from each value. The enhanced $H_2O_2$ generated by 5 μM $Ag^+$ and 20 μM ebselen-treated *E. coli* DHB4 cells were verified (***$P$ = 0.00057; Student's *t*-test).

Data information: In (B and C), data are presented as means ± SD of three independent experiments.

rare antibiotics that can currently combat them or that are being developed. We reported here that the synergistic bactericidal effect of $Ag^+$ with ebselen in combination was efficient against these MDR Gram-negative pathogens (Table 1). Further, results from animal experiments indicated that this new antibiotic combination should be considered as a candidate for clinical trials against MDR bacteria (Figs 6 and EV3, Table 1), and the system targeted by the combination is critical for bacterial survival, and thus, development of resistant mutants is not frequent as described by our previous study (Gustafsson *et al*, 2016).

The fact that $Ag^+$ and ebselen were efficiently against clinical isolates of MDR Gram-negative bacteria (Table 1), which indicated that they acted a mechanism that are different from existing antibiotics. The results we presented here proposed mechanisms for the synergistic antibacterial effect of $Ag^+$ with ebselen in combination. Silver and ebselen can directly inhibit *E. coli* TrxR, and fast deplete GSH, which resulted in the elevation of ROS production to determine cell death (synopsis). Thiol-dependent redox pathways regulate various central cellular functions. Thus, $Ag^+$ with ebselen in combination can react with SH-groups in

GSH, and particularly Trx and TrxR and possibly many other proteins, indicating that the inhibitory effect of $Ag^+$ with ebselen in combination may involve several cellular targets. In addition, $Ag^+$ and ebselen might target other molecules: For example, diguanylate cyclase and *M. tuberculosis* antigen 85 have been reported to be targets of ebselen (Favrot *et al*, 2013; Lieberman *et al*, 2014). This may impair the development of antibiotic resistance in bacteria.

Though most of the Gram-negative bacteria have both Trx and GSH/Grx systems as mammalian cells, and silver can also inhibitory activity against mammalian TrxR (Fig EV4), yet the combination of $Ag^+$ and ebselen exhibited selective synergistic toxicity on bacteria (Figs 1 and 6). The reasons for the selective toxicity of $Ag^+$ with ebselen in combination toward prokaryotic cells might be explained by: First, mammalian cells possess various tissue-specific GPxs, which contain Sec in their active site and make them highly efficient to remove hydrogen peroxide with a reaction rate of $10^8$ $M^{-1} \cdot s^{-1}$ (Papp *et al*, 2007; Lu & Holmgren, 2009, 2014). In contrast, only a Cys-containing GPx with a low ROS removing capacity is predicted to be present in bacteria (Lu & Holmgren, 2014). Second, ebselen is

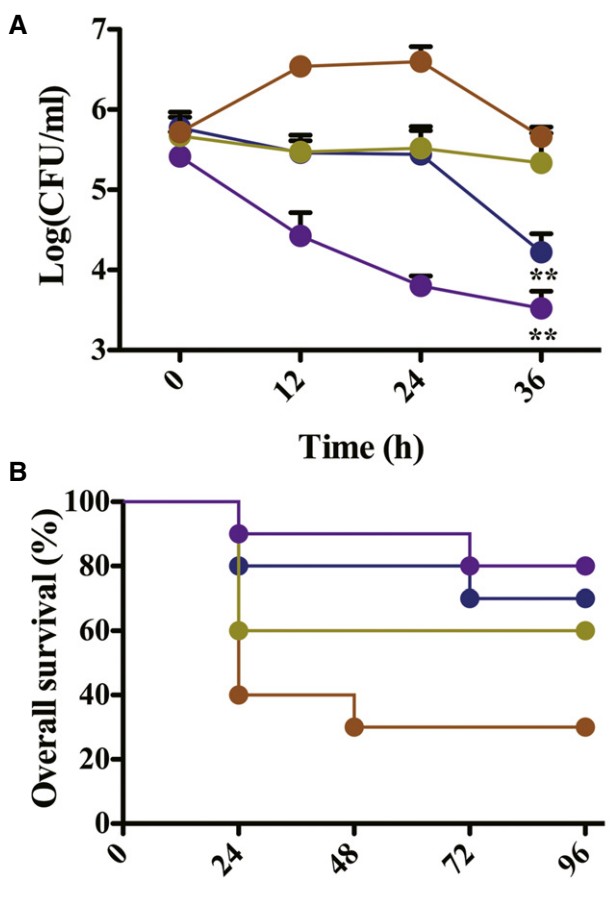

**A** Control

**Ebselen (25 mg/Kg body weight)**

**AgNO$_3$ (6 mg/Kg body weight)**

**AgNO$_3$ + Ebselen (6 + 25 mg/Kg body weight)**

**Figure 6.  Mode of action of silver and ebselen in *in vivo* mild and acute mice peritonitis model.**

A   Mild mice peritonitis model. Mice were infected by intraperitoneal administration of 100 μl of $1.7 \times 10^6$ *E. coli* ZY-1 cells. After 24 h, 12 mice per group received antibacterial treatments (25 mg ebselen/kg and 6 mg AgNO$_3$/kg body weight). 12, 24, and 36 h after treatment, the peritoneal fluid was collected for analysis of *E. coli* CFU (*n* = 12 mice for each group), and data are presented as means ± SD of three independent experiments. **$P < 0.01$ (Student's *t*-test).

B   Acute mice peritonitis model. Inoculation was performed by intraperitoneal injection of 100 μl of $6.0 \times 10^6$ CFU/ml *E. coli* ZY-1 inoculums. After inoculation for 1 h, 10 mice per group received antibacterial treatments, and the mice were observed for 7 days to evaluate overall survival (*n* = 10 mice for each group), and the experiment was performed in duplicate.

a substrate of the Sec-containing TrxR of mammalian cells, which makes ebselen act as an antioxidant by facilitating the electron transfer to reduce peroxides and peroxynitrite (Zhao & Holmgren, 2002). Meanwhile, ebselen is an irreversible inhibitor of bacterial TrxR and only blocks the electron transfer via TrxR. The ebselen

analog, ebsulfur, was toxic for the parasite *Trypanosoma brucei* and induced a high level of ROS to enhance the inhibition of trypanothione reductase (TryR) (Lu *et al*, 2013b). The ROS elevation caused by Ag$^+$ and ebselen described here may cause damage in a similar manner. The increased ROS levels in bacteria may also increase ebselen binding resulting in the inhibition of other sulfhydryl-dependent enzymes.

Previous work showed that silver can enhance antibacterial effects of classic antibiotics (Morones-Ramirez *et al*, 2013b), and more works reported that the lethality of current antibiotics was accompanied by redox physiology alteration and ROS generation (Dwyer *et al*, 2014; Belenky *et al*, 2015). Our work directly showed that silver with ebselen in combination could target redox system, and has a significant synergistic antibacterial effect on clinic important Gram-negative bacteria.

Current antibacterial strategies are predominantly based on inhibition of cell wall synthesis, inhibition of DNA and RNA synthesis and replication, and inhibition of protein synthesis. Redox system is a universal anti-oxidative system, which is essential for living organism; inhibition of redox system will result in oxidative stress, which showed a novel antibacterial principle that could be used to screen new antibiotics.

All in all, our results with silver and ebselen in synergistic combination implied that targeting bacterial thiol systems was a potent strategy against bacterial infections and a novel promising antibiotic mechanism.

## Materials and Methods

### Bacterial strains

All *in vitro* experiments were performed with *Escherichia coli* (*E. coli*) DHB4 and its derived redox phenotypes (Appendix Table S3), and clinically isolated multidrug-resistance (MDR) Gram-negative strains (Appendix Tables S1, S2 and S4). All *in vivo* experiments were performed with *E. coli* ZY-1 (Appendix Table S4), which was isolated from the urine of clinical patient in the First Affiliated Hospital of Three Gorges University in Hubei Province, P. R. China, with an approval for research from the Ethics Committee of First Affiliated Hospital of Three Gorges University and an informed-consent of the patient. The strain was thoroughly identified and stored in our laboratory. Other clinical isolated MDR Gram-negative strains (Appendix Tables S1 and S2) were obtained from clinical patients in Renmin Hospital of Three Gorges University in Hubei Province, PRC, with all approvals and informed consents.

### Antibiotics and chemicals

All experiments were performed in Luria-Bertani (LB) medium (EMD millipore). Unless otherwise specified, the following concentrations were used for the antibacterial experiments with *E. coli* strains and clinical pathogens: 0, 1, 2, 4, 5, 20, 40, 80 μM 2-phenyl-1,2-benzisoselenazol-3(2H)-one (ebselen) (Daiichi), and 0, 0.625, 1.25, 2.5, 5, 10, 20, 40, 80 μM silver nitrate (Sigma-Aldrich). 4-acetamido-4′-maleimidylstilbene-2,2′-disulfonic acid (AMS) (Invitrogen), protease inhibitor cocktails (Roche), DC™ protein assay (Bio-Rad), propidium iodide (PI) (BD Biosciences), *E. coli* DHB4 TrxR, sheep anti-*E. coli*

**Table 4. Analysis of blood samples from mice treatment with or without silver and ebselen.**

| | Vehicle | | | Ebselen + Ag | | | PBS | | |
|---|---|---|---|---|---|---|---|---|---|
| | 6 h | 24 h | 48 h | 6 h | 24 h | 48 h | 6 h | 24 h | 48 h |
| ALT (U/l) | 32.7 ± 9.3 | 24.0 ± 3.6 | 25.33 ± 4.0 | 79.0 ± 41.0 | 31.7 ± 6.4 | 25.8 ± 5.62 | 28.7 ± 6.4 | 25.3 ± 7.2 | 28.3 ± 17.0 |
| AST (U/l) | 130.7 ± 46.6 | 108.0 ± 5.0 | 106.67 ± 38.1 | 300.7 ± 15 | 145.3 ± 28.6 | 102.5 ± 53.4 | 113.3 ± 6.0 | 110.3 ± 23.5 | 88.7 ± 7.0 |
| BUN (mmol/l) | 5.3 ± 1.14 | 5.9 ± 0.93 | 6.74 ± 2.0 | 6.0 ± 0.71 | 6.53 ± 0.8 | 5.74 ± 0.5 | 4.6 ± 0.8 | 7.2 ± 0.6 | 5.8 ± 1.5 |
| CRE (μmol/l) | 29.3 ± 2.52 | 37.7 ± 1.53 | 29.7 ± 2.08 | 23.0 ± 1.7 | 39.0 ± 2.6 | 30.0 ± 5.5 | 23.0 ± 3.5 | 26.7 ± 2.3 | 28.3 ± 5.9 |
| TBIL (μmol/l) | 0.10 ± 0.17 | 0 | 0 | 0.43 ± 0.40 | 0 | 0 | 0.40 ± 0.5 | 0 | 0 |
| WBC ($10^9$/l) | 3.29 | 2.52 | 3.50 | 4.36 | 3.27 | 2.21 | 3.12 | 3.13 | 2.38 |
| Neu# ($10^9$/l) | 1.46 | 0.69 | 0.84 | 2.69 | 1.20 | 0.52 | 0.82 | 0.90 | 0.55 |
| Lym# ($10^9$/l) | 1.72 | 1.72 | 2.57 | 0.07 | 2.39 | 1.57 | 2.2 | 1.98 | 1.78 |
| Mon# ($10^9$/l) | 0.07 | 0.05 | 0.03 | 1.57 | 0.05 | 0.01 | 0.02 | 0.06 | 0.04 |
| Eos# ($10^9$/l) | 0.02 | 0.06 | 0.06 | 0.03 | 0.07 | 0.01 | 0.08 | 0.18 | 0.01 |
| Bas# ($10^9$/l) | 0.02 | 0 | 0 | 0 | 0.01 | 0 | 0 | 0.01 | 0 |
| IMG# ($10^9$/l) | 0 | 0 | 0.01 | 0.02 | 0.01 | 0 | 0 | 0.04 | 0 |
| Neu% (%) | 44.3 | 27.3 | 24.1 | 61.7 | 32.2 | 23.5 | 26.3 | 28.7 | 23.3 |
| Lym% (%) | 52.3 | 68.4 | 73.3 | 1.6 | 64.5 | 75.3 | 70.4 | 63.4 | 74.5 |
| Mon% (%) | 2 | 2 | 0.9 | 35.9 | 1.3 | 0.5 | 0.6 | 1.9 | 1.7 |
| Eos% (%) | 0.7 | 2.2 | 1.7 | 0.7 | 1.8 | 0.5 | 2.6 | 5.7 | 0.4 |
| Bas% (%) | 0.7 | 0.1 | 0 | 0.1 | 0.2 | 0.2 | 0.1 | 0.3 | 0.1 |
| IMG% (%) | 0 | 0.1 | 0.2 | 0.3 | 0.3 | 0.1 | 0 | 1.4 | 0 |
| PLT ($10^9$/l) | 574 | 912 | 956 | 566 | 602 | 744 | 605 | 187 | 928 |
| MPV (fl) | 7.7 | 6.7 | 6.5 | 7.3 | 7.1 | 6.1 | 7.0 | 7.9 | 6.1 |
| PDW | 14.8 | 14.8 | 14.7 | 14.9 | 15 | 14.7 | 15 | 15.4 | 14.6 |
| PCT (%) | 0.442 | 0.613 | 0.62 | 0.413 | 0.428 | 0.456 | 0.422 | 0.147 | 0.566 |
| P-LCC ($10^9$/l) | 76 | 72 | 65 | 66 | 61 | 39 | 62 | 37 | 48 |
| P-LCR (%) | 13.2 | 7.9 | 6.8 | 11.7 | 10.1 | 5.3 | 10.3 | 19.7 | 5.1 |
| RBC ($10^{12}$/l) | 9.26 | 8.56 | 8.35 | 7.95 | 7.25 | 8.21 | 8.31 | 8.91 | 7.79 |
| HGB (g/l) | 145 | 129 | 127 | 124 | 112 | 130 | 133 | 141 | 122 |
| HCT (%) | 48.7 | 43.3 | 42 | 41.8 | 38 | 44.8 | 44.6 | 46.6 | 41.2 |
| MCV (fl) | 52.7 | 50.6 | 50.3 | 52.5 | 52.4 | 54.5 | 53.7 | 52.3 | 52.8 |
| MCH (pg) | 15.77 | 15.1 | 15.2 | 15.6 | 15.5 | 15.9 | 16.0 | 15.8 | 15.6 |
| MCHC (g/l) | 297 | 299 | 301 | 298 | 296 | 291 | 297 | 302 | 296 |
| RDW-CV (%) | 18.1 | 18.2 | 18.1 | 20.2 | 16.3 | 21.6 | 17.8 | 21.2 | 18.1 |
| RDW-SD (fl) | 33.2 | 31.9 | 32.0 | 36.7 | 29.5 | 41.1 | 33.3 | 38.7 | 33.5 |

ALT, alanine transaminase; AST, aspartate aminotransferase; BUN, blood urea nitrogen; CRE, creatinine; TBIL, total bilirubin; WBC, white blood cell count; Neu, neutrophil; Lym, lymphocyte; Mon, monocyte; Eos, eosinophil; Bas, basophil; IMG, immunoglobulin; PLT, platelets; MPV, mean platelet volume; PDW, platelet distribution width; PCT, plateletcrit; P-LCC, large platelet count; P-LCR, large platelet ratio; RBC, red blood cell count; HGB, hemoglobin; HCT, hematocrit; MCV, mean corpuscular volume; MCH, mean corpuscular hemoglobin; MCHC, mean corpuscular hemoglobin concentration; RDW-CV, red cell distribution width coefficient of variation; RDW-SD, red cell distribution width standard deviation. Data are means ± SD of three independent experiments.

Trx1 antibody, and rabbit anti-*E. coli* Trx2 antibody were from IMCO Corp. (Stockholm, Sweden; http://www.imcocorp.se; Lu *et al*, 2013a), goat anti-rabbit IgG-HRP (Santa Cruz, lot# H1015; Lu *et al*, 2013a), rabbit anti-goat IgG-HRP (Southern Biotech, lot# 12011-PG56; Lu *et al*, 2013a), IgG2a mouse monoclonal antibody for glutathione–protein complexes (VIROGEN, lot# 101-A, clone number D8), 4–12% bolt Bis-Tris gel (VWR), all the other reagents were from Sigma-Aldrich.

## Synergistic effect of silver with ebselen in combination on *E. coli* growth

*Escherichia coli* DHB4 cells overnight cultures were diluted 1:1,000 times in Luria-Bertani (LB) medium and treated with serial concentrations of AgNO₃ and/or ebselen for 16 h. The cell viability was determined by measuring the absorbance at 600 nm. The culture treated with 0.8% (v/v) DMSO was used as a control.

## Toxicity analysis of silver with ebselen in combination against mammalian cells

HeLa cells were purchased from ATCC, and through mycoplasma detection and human cell line authentication by STR analysis (ATCC, U.S.A). HeLa cells cultured in DMEM medium supplemented with 10% FCS, 100 units/ml penicillin, and 100 μg/ml streptomycin at 37°C in a 5% $CO_2$ incubator. The cells were seeded in 96 micro-well plates and grown till 70–80% confluency. The cells were treated with serial combinations of ebselen and $AgNO_3$ for 24 h. The cell toxicity was detected by MTT assay (Lewis, 2013).

## Antibacterial effect of silver with ebselen in synergistic combination on the growth of clinical isolated MDR Gram-negative strains

Ten clinical isolated MDR Gram-negative (GSH-positive) strains were grown until an $OD_{600\ nm}$ of 0.4 and were diluted 1:100 into 100 μl of LB medium in 96 micro-well plates. Serial dilutions of 100 μl ebselen and $AgNO_3$ were added to the individual wells. The minimum inhibitory concentration (MIC) was determined after 16 h culture at 37°C. The culture treated with 0.8% (v/v) DMSO was used as a control.

## Detection of bactericidal effect of silver with ebselen in combination on *E. coli* strains

*Escherichia coli* DHB4 cells were grown in 15-ml tubes until an $OD_{600\ nm}$ of 0.4, and treated with 5 μM $AgNO_3$ and serial concentration of ebselen (0, 20, 40, 80 μM) in combination. The survival of untreated *E. coli* was compared with the antibiotic-treated cells by measuring $OD_{600\ nm}$ and counting the colonies. For colony formation assay, cells were harvested at 10 min, 1, 2 and 4 h by centrifugation at 7,080 *g* for 5 min and thoroughly washed three times with PBS. The cells were serially diluted in PBS, and 100-μl cultures were plated on LB plates. The colonies were counted after overnight incubation, and CFU/ml was calculated using the following formula: [(colonies)×(dilution factor)]/(amount plated).

Further, cells cultured and washed as above were harvested at 10 min by centrifugation at 6,000 rpm for 5 min and thoroughly washed three times with PBS. Nuclei were stained with 5 μg/ml propidium iodide (PI) for 20 min in the absence of a cell permeate and analyzed by flow cytometry (CyAn adp, Beckman coulter).

## Measurement of Trx/TrxR activity and GSH amount in silver and ebselen-treated *E. coli* cell lysates

*Escherichia coli* DHB4 cells were grown till the absorbance at $OD_{600\ nm}$ of 0.4 in LB medium, and the bacterial cells were treated with different dilutions of ebselen and $AgNO_3$ for 10 min. Cells were harvested by centrifugation at 6,000 rpm for 5 min and thoroughly washed three times with PBS, and then, cells were re-suspended in lysis buffer (25 mM Tris–HCl, pH 7.5, 100 mM NaCl, 2.5 mM EDTA, 2.5 mM EGTA, 20 mM NaF, 1 mM $Na_3VO_4$, 20 mM sodium β-glycerophosphate, 10 mM sodium pyrophosphate, 0.5% Triton X-100) containing protease inhibitor cocktail and lysed by sonication. The cell lysates were obtained by centrifugation at 15,340 *g* for 20 min, and the protein concentration was measured by Lowry protein assay (Bio-Rad DC™).

*Escherichia coli* DHB4 TrxR activity in cell extracts was measured by a DTNB reduction activity assay (Holmgren & Bjornstedt, 1995). The experiments were performed with 96 micro-well plates in the solution containing 50 mM Tris–HCl (pH 7.5), 200 μM NADPH, 1 mM EDTA, 1 mM DTNB, in the presence of 5 μM *E. coli* Trx. The absorbance at 412 nm was measured for 5 min with a VERSA micro-well plate reader, and the slope of initial 2 min was used to represent TrxR activity. The Trx activity was detected by this method coupled with 100 nM *E. coli* TrxR instead of 5 μM *E. coli* Trx in the reaction mixture.

To measure GSH levels, 25 μg of the cell lysates was added in the solution containing 50 nM GR, 50 mM Tris–HCl (pH 7.5), 200 μM NADPH, 1 mM EDTA, 1 mM DTNB. The absorbance at 412 nm was measured for 5 min.

## Trx redox state in *E. coli* treated with silver and ebselen in combination

*Escherichia coli* DHB4 cells were grown till the absorbance at $OD_{600\ nm}$ of 0.4 in LB medium, and the bacterial cells were treated with different dilutions of ebselen and $AgNO_3$ for 10 min. Western blotting was performed to detect the Trx1 and Trx2 redox state of the ebselen and $AgNO_3$-treated *E. coli* cells. The cells were harvested by centrifugation at 6,000 rpm for 5 min and thoroughly washed three times with PBS, and precipitated the protein with 5% TCA in 1.0 ml. The precipitates were washed with 1 ml pre-ice-cold acetone for three times and dissolved in 50 mM Tris–HCl (pH 8.5) with 0.5% SDS containing 15 mM AMS at 37°C for 2 h. Proteins were obtained by centrifugation at 13,000 rpm for 20 min to remove the pellets, and the protein concentration was measured by Lowry protein assay (Bio-Rad DC™). Redox state of Trx1 and Trx2 was detected with sheep anti-*E. coli* Trx1 antibody and (Rabbit anti-*E. coli* Trx2 antibody) at 1:1,000 dilution, followed by the detection of Chemiluminescence Reagent Plus.

## Proteins S-glutathionylation in *E. coli* treated with silver and ebselen in combination

Total proteins S-glutathionylation of the ebselen with $AgNO_3$ in combination-treated *E. coli* cells were detected by Western blotting. Cells were cultured and washed as described above, and re-suspended in lysis buffer (25 mM Tris–HCl, pH 7.5, 100 mM NaCl, 2.5 mM EDTA, 2.5 mM EGTA, 20 mM NaF, 1 mM $Na_3VO_4$, 20 mM sodium β-glycerophosphate, 10 mM sodium pyrophosphate, 0.5% Triton X-100, protease inhibitor cocktail) containing 30 mM IAM. After lysed by sonication, the cell lysates were obtained by centrifugation at 13,000 rpm for 20 min. Protein concentration was measured by Lowry protein assay (Bio-Rad DC™). Samples were incubated with SDS-loading buffer at 90°C for 10 min and then separated on the 4–12% bolt Bis-Tris gel with MES running buffer (150 V, 40 min). Western blotting assay was performed with IgG2a mouse monoclonal antibody (VIROGEN, 101-A/D8) for glutathione–protein complexes.

## Synergistic effect of silver and ebselen on the growth of *E. coli* DHB4 redox phenotypes

Eleven *E. coli* DHB4 redox phenotypes were grown until an $OD_{600\ nm}$ of 0.4 and were diluted 1:100 into 100 μl of LB medium in

96 micro-well plates. Serial dilutions of ebselen and AgNO$_3$ were added to the individual wells. The minimum inhibitory concentration (MIC) was determined after 24 h culture at 37°C. The culture treated with 0.8% (v/v) DMSO was used as a control.

### Inhibition of recombinant bacterial Trx/TrxR by silver

Inhibition of recombinant bacterial TrxR by silver was performed by using *E. coli* enzyme with the method previously described (Holmgren & Bjornstedt, 1995). The experiments were performed with 96 micro-well plates in the solution containing 50 mM Tris–HCl (pH 7.5), 200 μM NADPH, 1 mM EDTA, 1 mM DTNB, in the presence of 5 μM *E. coli* Trx. The absorbance at 412 nm was measured for 5 min with a VERSA micro-well plate reader, and the slope of initial 2 min was used to represent TrxR activity. The Trx activity was detected by this method, coupled with the use of 100 nM *E. coli* TrxR instead of 5 μM *E. coli* Trx in the reaction mixture.

### Analysis of fluorescent spectra

Fluorescent spectra of reduced *E. coli* Trx with silver were recorded at 10 μM in a PerkinElmer Enspire multilabel recorder using an excitation at 280 nm (Holmgren & Bjornstedt, 1995).

### Measurement of ROS production

*E. coli* DHB4 cells were grown till the absorbance at OD$_{600 \text{ nm}}$ of 0.4 in LB medium, and the bacterial cells were treated with different combinations of ebselen and AgNO$_3$ for 10 min. To analyze the amount of ROS production in the bacteria, cells were harvested by centrifugation at 6,000 rpm for 5 min and thoroughly washed three times with PBS, and stained with 5 μM H$_2$DCF-DA for 20 min. After the incubation, cells were spun down and re-suspended in PBS, and the ROS production was quantified by flow cytometry (CyAn adp, Beckman coulter).

### H$_2$O$_2$ production

*E. coli* DHB4 cells were grown till the absorbance at OD$_{600 \text{ nm}}$ of 0.4 in LB medium, and the bacterial cells were treated with 20 μM ebselen and 5 μM AgNO$_3$ for 10 min. Cells were harvested by centrifugation at 6,000 rpm for 5 min and thoroughly washed three times with PBS, and sonicated for 10 s. In the presence of 50 μM Amplex$^®$ Red reagent, 0.1 U/ml HRP in 50 mM sodium phosphate buffer, pH 7.4, 50 μl of samples was incubated for 30 min at room temperature protected from light, and detected with absorbance at 560 nm (Molecular Probes, Eugene, OR).

### Mild peritonitis mice model assay

Approval from the Medical Animal Care & Welfare Committee of China Three Gorges University was obtained prior to using the animals for research. Healthy 6-week-old Kunming male mice (body weight 18 ± 2 g) were purchased from Laboratory Animal Center of China, Three Gorges University. All mice were kept in individually ventilated cages (five mice per cage) under a constant dark (12 h)–light (12 h) cycle in a conventional SPF animal house and were free access to food and water. Five mice were sampled

**The paper explained**

**Problem**

Antibiotic resistance has become a great worldwide challenge. Gram-negative bacteria present a major threat to human life and medicine, with almost no efficacious antibiotics available for treatment making it urgent to find new principles and mechanisms.

**Results**

Here we found that silver acts synergistically with the selenazol drug ebselen to combat five clinically most difficult-to-treat multidrug-resistant Gram-negative bacteria, by targeting thiol-dependent antioxidant systems. The results were further validated by successfully treating mice with multidrug-resistant *E. coli*-caused mild or acute peritonitis.

**Impact**

Our research presents a new strategy for the development of antibiotics against multidrug-resistant Gram-negative bacteria.

randomly to examine bacterial recovery from the brain, liver, spleen, and kidney to rule out *E. coli* infection before experimental manipulation, and no bacteria were detected.

The experimentation was performed in random block design and single-blind trial, and the sample size was determined as described by Krzywinski M (Krzywinski & Altman, 2013). The sample size was calculated by power analysis and estimated as: corrected sample size = sample size/(1–[% attrition/100]). Forty-eight mice were divided into four groups, 12 mice/group. Inoculation was performed by intraperitoneal injection of 100 μl 1.7 × 10$^6$ *E. coli* ZY-1 cells using a 26-gauge syringe. The inoculum was delivered in suspension with 8% (w/v) mucin in sterile saline. Twenty-four hours after introduction of the inoculum, 12 mice per group received antibacterial treatments. 0, 12, 24, and 36 h post-infection, peritoneal washes were performed by injecting 1.0 ml of sterile saline in the intraperitoneal cavity followed by a massage of the abdomen (100 times/mouse). Subsequently, the abdomen was opened and 200 μl of peritoneal fluid (PF) was recovered from the peritoneum for analysis of *E. coli* CFU/ml, and CFU/ml was enumerated. For the CFU/ml measurement, the peritoneal fluid was serially diluted in PBS (pH 7.6). Experiments are performed triplicate.

### Acute peritonitis mice model assay

The experimentation was designed in random block design and single-blind trial, and 40 mice were divided into four groups, 10 mice/group. Inoculation was performed by intraperitoneal injection of 100 μl of 6.0 × 10$^6$ CFU/ml *E. coli* ZY-1 inoculums using a 26-gauge syringe. The inoculum was delivered in suspension with 8% (w/v) mucin in sterile saline. One hour after introduction of the inoculum, 10 mice per group received antibacterial treatments, and the mice were observed for 7 days to evaluate overall survival. Experiments were performed in duplicate.

### *In vivo* toxicity analysis of silver with ebselen in combination

Five mice per group were treated with 6 mg AgNO$_3$/kg body weight and serial concentration of ebselen (10, 15, 20, 25 mg AgNO$_3$/kg

body weight) intraperitoneally. Mice were observed for 7 days, and the overall survival was calculated.

### Blood samples analysis

Three mice per group were treated with parenterally administered PBS, 25 mg ebselen/kg body weight in combination with 6 mg $AgNO_3$/kg body weight, and vehicle. The animals were observed for 2 days, and retro-orbital blood sample collection was performed 6, 24, and 48 h after treatment. Blood was collected in heparinized whole blood test tubes and further analyzed by Blood Chemistry Analyzer (SYSMEX XE5000).

### Effect of ebselen on *E. coli* growth

*Escherichia coli* DHB4 cells were grown until an $OD_{600\ nm}$ of 0.4 and treated with serial dilutions of ebselen (0, 2, 4, 8 μM) for 16 h. The cell viability was determined by measuring the absorbance at 600 nm. The culture treated with 0.8% (v/v) DMSO was used as a control.

### Direct survival rate assay

The direct survival rate assay was performed to assess the survival capacity of ebselen and $AgNO_3$-treated *E. coli* DHB4 strain in healthy mice blood. The phosphate-buffered saline (PBS, pH 7.6)-treated cells were used as the positive control, and the experiment was performed in duplicate. Briefly, blood was extracted from three mice and collected in heparinized tubes. Approximately 100 *E. coli* DHB4 cells were harvested during the logarithmic phase, washed with sterile PBS, and added to 100 μl blood. After incubation at 37°C for 6 h, duplicate 100-μl aliquots from each blood sample were spread onto LB agar, and the surviving colonies were enumerated after overnight incubation. The results showed that ebselen and $AgNO_3$ could help innate immunity to clear *E. coli*.

### Inhibition of recombinant mammalian TrxR by silver

The experiments were performed with 96 micro-well plates in the solution containing 50 mM Tris–HCl (pH 7.5), 250 μM NADPH, 1 mM EDTA, 1 mM DTNB, in the presence of 10 nM *E. coli* TrxR. The absorbance at 412 nm was measured for 5 min with a VERSA micro-well plate reader, and the slope of initial 2 min was used to represent TrxR activity.

### Statistical analysis

Mean, standard deviation (SD), and *t*-test (two-tailed, unpaired) significances were calculated in GrapPad Prism Software. *$P < 0.05$, **$P < 0.01$, ***$P < 0.001$.

Expanded View for this article is available online.

### Acknowledgements

This work was supported by the Swedish Cancer Society (961), the Swedish Research Council Medicine (13X-3529), the K&A Wallenberg Foundation, and grants from Karolinska Institutet, National Natural Science Foundation of China NO. 81550028 to J.W.

### Author contributions

JL, LZo, and AH conceived the study. JL, LZo, JW, and AH designed experiments. LZo, JL, JW, XR, LZh, and YG performed experiments. JL, LZo, MER, and AH wrote the paper. All authors analyzed the results and approved the final version of the manuscript.

### Conflict of interest

The authors declare that they have no conflict of interest. AH and JL have applied for a patent.

### For more information

http://ki.se/people/arnhol

http://pharmacy.swu.edu.cn/viscms/pharidex/bumen21044/20130805/310807.html

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
