## [Review Process File · EMBO Molecular Medicine]

Synergistic antibacterial effect of silver and ebselen against multidrug-resistant Gram-negative bacterial infections

Lili Zou, Jun Lu, Jun Wang, Xiaoyuan Ren, Lanlan Zhang, Yu Gao, Martin E Rottenberg, Arne Holmgren

Corresponding author: Jun Lu & Arne Holmgren, Karolinska Institute

Review timeline:

Submission date:	06 February 2017
Editorial Decision:	17 March 2017
Revision received:	19 April 2017
Editorial Decision:	03 May 2017
Revision received:	11 May 2017

Transaction Report:

Editor: Roberto Buccione

1st Editorial Decision

17 March 2017

Thank you for the submission of your manuscript to EMBO Molecular Medicine. We are very sorry that it has taken so long to get back to you on your manuscript.

In this case we experienced unusual difficulties in securing three willing and appropriate reviewers. Further to this, reviewer #1 ultimately failed to deliver his/her report notwithstanding repeated chasers and his/her assurances that s/he would deliver. As a further delay cannot be justified I have decided to proceed based on the two available consistent evaluations.

As you will see, both Reviewers find the study of interest and worthy of publication although reviewer 2 is more reserved and expresses a number of concerns focusing on both poor presentation of some of the experimental findings and lack of sufficient support for some of the conclusions. In aggregate, the points they raise appear well-taken and addressable within a reasonable timeframe.

While publication of the paper cannot be considered at this stage, we would be pleased to consider a revised submission, with the understanding that the Reviewers' concerns must be fully addressed including with additional experimental data where appropriate and that acceptance of the manuscript will entail a second round of review.

Please note that it is EMBO Molecular Medicine policy to allow a single round of revision only and that, therefore, acceptance or rejection of the manuscript will depend on the completeness of your responses included in the next, final version of the manuscript.

As you know, EMBO Molecular Medicine has a "scooping protection" policy, whereby similar findings that are published by others during review or revision are not a criterion for rejection.

However, I do ask you to get in touch with us after three months if you have not completed your revision, to update us on the status. Please also contact us as soon as possible if similar work is published elsewhere.

Please note that EMBO Molecular Medicine now requires a complete author checklist (<http://embomolmed.embopress.org/authorguide#editorial3>) to be submitted with all revised manuscripts. Provision of the author checklist is mandatory at revision stage; The checklist is designed to enhance and standardize reporting of key information in research papers and to support reanalysis and repetition of experiments by the community. The list covers key information for figure panels and captions and focuses on statistics, the reporting of reagents, animal models and human subject-derived data, as well as guidance to optimise data accessibility. In this case, the author checklist is especially relevant as, in addition to the concerns on the clinical features of the TMA, I note that both reviewers have reservations on your presentation of statistics information. The Author checklist will be published alongside the paper, in case of acceptance, within the transparent review process file

Please carefully adhere to our guidelines for authors (<http://embomolmed.embopress.org/authorguide>) to accelerate manuscript processing in case of acceptance.

I look forward to seeing a revised form of your manuscript as soon as possible.

***** Reviewer's comments *****

Referee #2 (Remarks):

This study aims at assessing the capacity of combining ebsalen and silver to fight multi-resistant strains of gram-negative bacteria. The study shows such combination could be usefull and aims at deciphering the molecular mechanism underlying the ebsalen potentialing effect. However, there are several issues, both scientific and editorial, that need to be clarified and be modified before any decision be taken.

1. Fig. 1 and page 5. There is no way one can figure out the MIC values (42 microM) and 3.2 microM from the Fig1A as indicated in the text. Please improve the presentation.
 2. Fig. 1 and page 5. Growth curves cannot be used to differentiate -cidal and -statics effect of drug.
 3. Fig. 1 and page 5. The combination ebselen 80 microM + Ag 5 microM is indeed bactericidal but it would have been nice to show the data with ebselen (at this concentration) alone.
 4. Page 6. Paragraph "Clinically isolated...". These clinical isolates are presumably GSH+. Please mention it explicitly.
 5. Page 6. same paragraph. Tables 2 and 3 are not showing what the text says they do. Please modify.
 6. Figure 3. Page 7. Data for Trx2 are not shown. Please do.
 7. Page 8. Paragraph "Silver irreversibly...". Would it be possible to have results using eucaryotic TrxR and Trx proteins here ? The whole paper rests on the differential sensitivity between human and bacterial enzymes but there is nothing here to substantiate this in the present work. For instance, the authors could use a human Trx/TrxR protein expressed in E coli and be subjected to the same treatment in parallel to the E coli one.
 8. Page 9. The issue on ROS needs clarification, additional experiment, further considerations and alternative interpretations : (i) The use of NAC is ambiguous; It could also protect against silver by scavenging it, rather than scavenging ROS. (ii) the use of dyes to follow ROS production has been largely discussed and questioned in the recent literature and this should be taken into account in the present study. In partiucular even if one trusts dyes and ROS concentration is indeed enhanced, the effect of such increase in cell viability remains to be tested, and the sentence "these results demonstrated..." be tuned down. (iii) were ROS to be involved one would expect MIC values of the oxyR mutant be different from the wt, whihc is not the case (Table 4). (iv) I could not find citation of the Science papers by the Lewis and Imlay labs in the text although they appear in the Literature section.
- Table 4. Could the Ebselen MIC values be given ?
 Fig EV2. E. coli is ZY-1 in the legend and DHB4 on the figure.
 Fig EV3. Unclear. 5 different concentrations, 3 curves visible

Fig EV4. Unclear. 5 different concentrations, 1 curve visible

Fig EV5. This model needs a legend and/or some associated text.

Referee #3 (Comments on Novelty/Model System):

The model systems were used for adequate for addressing the considered hypotheses.

Referee #3 (Remarks):

In this extensive and exciting paper the authors show that silver and ebselen can act synergistically against multidrug resistant bacteria. The study is well designed, rigorous, and very well executed. The authors work out associated mechanisms. This piece will generate considerable and discussion particularly given the growing need for new means to treat resistant infections. Before publishing this work the authors should address the following points:

- a. The authors should include a schematic illustrating the proposed mechanism(s) for the reported synergistic effects.
- b. The authors should note prior work reporting on antibacterial-induced ROS including some of the papers cited in the manuscript as well as Dwyer et al PNAS (2014) and Belenky et al Cell Reports (2015).
- c. The authors should expand the discussion as to how the reported insights could be utilized to create adjuvants for current antibiotics and new antibacterial strategies altogether.
- d. The authors should also expand the discussion on the clinical implications and feasibility of implementing new treatment modalities based on their discoveries.

1st Revision - authors' response

19 April 2017

Referee #2 (Remarks):

This study aims at assessing the capacity of combining ebsalen and silver to fight multi-resistant strains of gram-negative bacteria. The study shows such combination could be usefull and aims at deciphering the molecular mechanism underlying the ebsalen potentialing effect. However, there are several issues, both scientific and editorial, that need to be clarified and be modified before any decision be taken.

1. Fig. 1 and page 5. There is no way one can figure out the MIC values (42 microM) and 3.2 microM from the Fig.1A as indicated in the text. Please improve the presentation.

Thanks for the reviewer's suggestion. We have changed the presentation of Fig. 1. We changed the x-axis to the logarithm to base 2 (Log₂) instead of the logarithm to base 10 (Log₁₀, the common logarithm). The description of sentences has also been changed as "Ag⁺ alone inhibited *E. coli* growth with a minimal inhibition concentration (MIC) of 42 μM after 16 h treatment, while the addition of 2 μM ebselen dramatically decreased the MIC of Ag⁺ to 4.2 μM ($p=0.000028<0.001$) (Fig. 1A)." (page 5, lines 95-97, highlighted in blue).

2. Fig. 1 and page 5. Growth curves cannot be used to differentiate -cidal and -statics effect of drug. **Yes, we are agreed. In fact, we first detected the inhibition effect of silver with ebselen in combination on *E. coli* growth. As you can see in Fig. 2A, silver with ebselen in combination inhibited DHB4 growth (Fig. 2A). Further, we plated bacterial cultures from different time point, and the results showed that bacterial cultures from 5 μM silver and 80 μM ebselen treatment have little colony growth after 6 hours treatment (Fig. 2B). In order to clearly express the results, we state as "The growth curves showed a synergistic bacteriostatic effect of Ag⁺ with ebselen in combination in LB medium (Fig. 2A), and the synergistic bactericidal effect of 5 μM Ag⁺ and 80 μM ebselen in combination was further confirmed by the colony formation assay on LB-agar plates (Fig. 2B)" (page 5, lines 106-109, highlighted in blue).**

3. Fig. 1 and page 5. The combination ebselen 80 microM + Ag 5 microM is indeed bactericidal but it would have been nice to show the data with ebselen (at this concentration) alone.

Thank you for your advice. In fact, we detected the antibacterial effects of 0/20/40/80 μM ebselen on *E. coli* DHB4, and the results showed that in the first 8 h, only 80 μM ebselen showed inhibition effect on *E. coli* growth, but the cell viability gains back to the normal 12 h post-treatment. Since this paper focus on the synergistic antibacterial effect of silver and ebselen combination, in order to sustain the integrity, we add related results as expanded view figure (Fig. EV2, page 5, lines 110-111).

4. Page 6. Paragraph "Clinically isolated...". These clinical isolates are presumably GSH+. Please mention it explicitly.

Thank you for your suggestion. We add one sentence to emphasize all the clinical isolates we used here are GSH-positive bacteria, and state as "There are five most difficult-to-treat MDR Gram-negative pathogen species in the clinic, which are also typical GSH positive bacteria: *Klebsiella pneumonia*, *Acinetobacter baumannii*, *Pseudomonas aeruginosa*, *Enterobater cloacae* and *Escherichia coli*." (page 4, lines 80-81; page 6, lines 120-122, highlighted in blue).

5. Page 6. same paragraph. Tables 2 and 3 are not showing what the text says they do. Please modify.

Sorry for the bad presentation. We rewrote these sentences as "The isolated imipenem, cefepime, and cefotaxime-resistant *A. baumannii* (AB-1/2) and *E. cloacae* (ECL-1) strains were identified (Table 2, and 3), and were sensitive to Ag^+ with ebselen in combination (Table 1)" (page 6, lines 129-131).

6. Figure 3. Page 7. Data for Trx2 are not shown. Please do.

The reasons we are not showing the redox state of Trx2 is that Trx2 antigen epitopes can bind to acetylation reagent such as AMS, which makes Trx2 antibody functional incapacitated. Since only the oxidized Trx2 can be detected, diamide-oxidized Trx2 was used as a control to compare oxidize form of Trx2. As you can see in figure 3D, after parallel treatment as Trx1, the redox state of Trx2 showed no obvious change. Base on the results we obtained, Trx2 is less sensitive to the silver and ebselen in combination compared to Trx1. We added related results to Fig. 3, and main text (page 7, lines 140-149, highlighted in blue).

7. Page 8. Paragraph "Silver irreversibly..". Would it be possible to have results using eucaryotic TrxR and Trx proteins here? The whole paper rests on the differential sensitivity between human and bacterial enzymes but there is nothing here to substantiate this in the present work. For instance, the authors could use a human Trx/TrxR protein expressed in *E. coli* and be subjected to the same treatment in parallel to the *E. coli* one.

We appreciated the generosity of Dr. Cheng Qing who works in Karolinska Institute, for providing us highly purified human TrxR to perform the parallel experiments (Cheng Q, *et al.* Selenocysteine insertion at a predefined UAG codon in a release factor 1 (RF1) depleted *Escherichia coli* host strain bypasses species barriers in recombinant selenoprotein translation. *J Biol Chem.* 2017, doi: 10. 1074.). As you can see from the figure EV4, 5 nM Ag ion can inhibit human TrxR1 activity, which is the same when we used purified rat TrxR1 instead of human TrxR1. The inhibition effect of silver on purified rat TrxR1 was also shown by Srivastava M *et al.* (Srivastava S, Singh S, Self WT. *Environmental Health Perspectives.* 2012, 120(1): 56-61.). All the results showed that if we only consider silver ion, it can inhibit both bacterial and mammalian TrxR1 (Fig. EV4., and page 12, line 257).

In addition, our previous work showed that ebselen is a competitive inhibitor of *E. coli* TrxR, and a substrate of mammalian TrxR (Lu J, *et al.* Inhibition of bacterial thioredoxin reductase: an antibiotic mechanism targeting bacteria lacking glutathione. *FASEB J.* 2013, 27(4): 1394-1403.). This is the basis that we hypothesized redox system might be an antibacterial target. In this paper, we can see the selective synergistic toxicity of silver with ebselen in combination against bacteria over mammalian cells (page 4, lines 86-88). Combination of 2.5 μM ebselen and 5.0 μM silver could inhibit the growth of *E. coli* DHB4, and this combination showed no statistically significant effect on mammalian cells viability (Fig. 1, and page 5, lines 95-98). Meanwhile, 6 mg/kg body weight silver and 25 mg/kg body weight ebselen in combination showed no toxicity *in vivo* (Table 8), and the combination treatment can protect mice from mild and acute MDR *E. coli* peritonitis (Fig. 6). The results we obtained so far could not

elucidate the total complicated mechanism, but we believe that this specificity must be based on a more sophisticated highly efficiency anti-oxidant system including peroxiredoxins, catalase, glutathione peroxidases, etc. which mammalian cells possesses. We have described the reason of selectivity in pages 13-14, line 256-272, "The reasons for the selective toxicity of Ag^+ and ebselen in combination towards prokaryotic cells might be explained by: ...". Besides, we think silver and ebselen in combination act as the probe showing that redox system can work as an antibiotic target.

8. Page 9. The issue on ROS needs clarification, additional experiment, further considerations and alternative interpretations: (i) The use of NAC is ambiguous; It could also protect against silver by scavenging it, rather than scavenging ROS.

Yes, we agree with the reviewer's comments. When we design this experiment, NAC was considered as an efficient antioxidant agent. In order to verify whether NAC can directly scavenge Ag ion or not, the DTNB assay was performed, and the results showed that NAC can directly bind to silver. Further, we use GSH (1 mM) instead of NAC (2 mM) to perform the same experiment, the results showed that GSH also can directly bind to silver, which means that thiols can bind to silver. In all, NAC rescue experiments are not well-designed, and we decide to remove the related results.

Figure Inhibitory effects of silver on thiol.

2-200 μM NAC or GSH were incubated with 2-200 μM AgNO_3 solution, and then their activities were detected by DTNB reduction assay. The t-test significances were calculated between control and rest groups, and *: $p < 0.05$, **: $p < 0.01$, ***: $p < 0.001$.

(ii) the use of dyes to follow ROS production has been largely discussed and questioned in the recent literature and this should be taken into account in the present study. In particular even if one trusts dyes and ROS concentration is indeed enhanced, the effect of such increase in cell viability remains to be tested, and the sentence 'these results demonstrated...' be tuned down.

Yes, we agreed that $\text{H}_2\text{DCF-DA}$ method in ROS production has been under numerous discussions. Though it is not a perfect method, it is a widely used and accepted way to measure ROS production (West AP, *et al.* TLR signaling augments macrophage bactericidal activity through mitochondrial ROS. *Nature*, 2011, 472(7344): 476-480; Dwyer DJ, *et al.* Antibiotics induce redox-related physiological alterations as part of their lethality. *PNAS*, 2014, 111(20): 2100-2109.). It might not be quantitative; it could be qualitative (or semi-quantitative). Our results showed that silver with ebselen in combination targeted redox system, which are the two major antioxidant systems in bacteria. The elevated ROS production is 50 times higher than the control, which further testified the effect of silver with ebselen in combination. In order to further verify ROS production results we already obtained, we semi-quantified H_2O_2 level by Amplex Red, which is widely used to measure H_2O_2 in biological systems (Miwa S, *et al.* Carboxylesterase converts Amplex Red to resorufin: implications for mitochondrial H_2O_2 release assays. 2016, 90: 173-183; Dwyer DJ, *et al.* Antibiotics induce redox-related physiological alterations as part of their lethality. *PNAS*, 2014, 111(20): 2100-2109.), and the results showed that 5 μM Ag^+ and 20 μM ebselen in combination can cause significant H_2O_2 production compared to control (Fig. 5C, and page 9, lines 192-197, highlighted in blue). In

addition, *E. coli* mutants lacking OxyR components (*OxyR*), the H₂O₂ sensor which regulate the transcription of antioxidant genes in *E. coli*, were more sensitive to Ag⁺ and ebselen treatment compared to the wild type (WT) (Table 4-6). All results showed that lethality of Ag⁺ with ebselen against bacteria is accompanied by ROS generation.

Figure-5C Detection of H₂O₂ using the Amplex® Red Hydrogen Peroxide/Peroxidase Assay Kit (Invitrogen).

Reactions containing 50 µM Amplex® Red reagent, 0.1 U/mL HRP and the indicated amount of H₂O₂ in 50 mM sodium phosphate buffer, pH 7.4, were incubated for 30 minutes at room temperature, and detected with absorbance at 560 nm. Background determined for a non-H₂O₂ control reaction, has been subtracted from each value.

(iii) were ROS to be involved one would expect MIC values of the *oxyR* mutant be different from the wt, which is not the case (Table 4).

Yes, we agree that this result is weird, since OxyR can protect *E. coli* dehydratase clusters from H₂O₂ injury. If we check the MIC of ebselen in the presence of silver against *E. coli* DHB4 mutants, it seems more sensitive than the wild type. So, we double checked our record and repeated the experiments, and the results showed that 40 µM ebselen (Table 5) or 20 µM silver (Table 4) could inhibit *OxyR* mutant, which is indeed more sensitive than WT (page 9, lines 194-197).

(iv) I could not find citation of the Science papers by the Lewis and Imlay labs in the text although they appear in the Literature section.

Thank you very much. We updated endnote references library, and those two paper are excluded automatically.

Table 4. Could the Ebselen MIC values be given?

Yes, and the results are listed in Table 5.

Fig EV2. *E. coli* is ZY-1 in the legend and DHB4 on the figure.

Sorry for our carelessness, it is *E. coli* DHB4, ZY-1 is an *E. coli* strain that we used in other experiments, such as animal experiments. We already changed it (Fig. EV3, page 39, lines 792, highlighted in blue).

Fig EV3. Unclear. 5 different concentrations, 3 curves visible.

The fact is that three curves overlapped. Since this figure is not essential and due to the limitation of expanded figures number, we deleted this figure, and only described in the main text (page 10, line 203, highlighted in blue).

Fig EV4. Unclear. 5 different concentrations, 1 curve visible

The fact is that all the curves overlapped. 6 mg AgNO₃/kg body weight and a serial concentration of ebselen could not cause any death, which remained 100% survivals (five curves overlapped). Since this figure is not essential and the limitation of expanded figures number, we deleted this figure, and only described in the main text (page 10, lines 216-217, highlighted in blue).

Fig EV5. This model needs a legend and/or some associated text.

Yes, exactly. This is a schematic summary which proposed mechanisms for the reported synergistic effect of silver with ebselen in combination. We use it as synopsis.

Referee #3 (Comments on Novelty/Model System):

The model systems were used for adequate for addressing the considered hypotheses.

Referee #3 (Remarks):

In this extensive and exciting paper the authors show that silver and ebselen can act synergistically against multidrug resistant bacteria. The study is well designed, rigorous, and very well executed. The authors work out associated mechanisms. This piece will generate considerable and discussion particularly given the growing need for new means to treat resistant infections. Before publishing this work the authors should address the following points:

a. The authors should include a schematic illustrating the proposed mechanism(s) for the reported synergistic effects.

Thank you for your suggestion, we have included a schematic abstract as synopsis.

A. Silver ions were strong inhibitors of both *E. coli* thioredoxin (Trx) and thioredoxin reductase (TrxR).

B. Our previous results showed that ebselen is a substrate of mammalian TrxR but a competitive inhibitor of bacterial TrxR.

C. Silver and ebselen can directly inhibit *E. coli* TrxR, and fast deplete GSH, which resulted in elevation of ROS production to determine cell death, which should be considered as a candidate for clinical trials against MDR bacteria, and the system targeted by the combination are critical for bacterial survival and thus development of resistant mutants is not frequent as described by our previous study.

D. Silver and ebselen act as a probe to target essential bacterial systems which might be developed for novel efficient treatments against MDR Gram-negative bacterial infections.

Information: Trx, thioredoxin; TrxR, thioredoxin reductase; GSH, glutathione; GR, glutathione reductase; Grx: glutaredoxin; Prx, peroxiredoxins; MSR, methionine sulfoxide reductase; RNR, ribonucleotide reductase; ROS, reactive oxygen species.

b. The authors should note prior work reporting on antibacterial-induced ROS including some of the papers cited in the manuscript as well as Dwyer et al PNAS (2014) and Belenky et al Cell Reports (2015).

Thank you very much, these two articles are very good and can support our hypothesis and results, which certainly should be cited (page 13, lines 273-276). In Dwyer's paper, they found that antibiotics altered bacterial redox physiology, and the lethality was accompanied by ROS generation. In Belenky's paper, they profiled the *E. coli* metabolome and showed that antibiotics elevated redox state, and increased oxidative stress by tightly regulated glutathione pools.

c. The authors should expand the discussion as to how the reported insights could be utilized to create adjuvants for current antibiotics and new antibacterial strategies altogether.

Thank you for your advice. Previous work showed that silver can enhance antibacterial effects of classic antibiotics (Morones-Ramirez et al, 2013b), and more works reported that the lethality of current antibiotics was accompanied by redox physiology alteration and ROS generation (Dwyer et al, 2014) (Belenky et al, 2015). Our work directly showed that silver and ebselen in combination could target redox system, and has a significant synergistic antibacterial effect on Gram-negative bacteria (page 13, lines 273-278, highlighted in blue).

d. The authors should also expand the discussion on the clinical implications and feasibility of implementing new treatment modalities based on their discoveries.

Current antibacterial strategies are predominantly based on inhibition of cell wall synthesis, inhibition of DNA and RNA synthesis and replication, and inhibition of protein synthesis. Redox system is a universal anti-oxidative system which is essential for living organism, inhibition of redox system will result in oxidative stress, which showed a novel antibacterial principle that could be used to screen new antibiotics (page 13, lines 279-283, highlighted in blue).

2nd Editorial Decision

03 May 2017

Thank you for the submission of your revised manuscript to EMBO Molecular Medicine. We have now received the enclosed reports from the referees that were asked to re-assess it. As you will see the reviewers are now globally supportive and I am pleased to inform you that we will be able to accept your manuscript pending the following final amendments:

- 1) Reviewer 1 notes that the text requires some work to improve readability and clarity. We agree (including on the passage mentioned by the reviewer. I have also taken the liberty to suggest some modifications to the Title, Abstract and "The Paper Explained" sections. Please accept (or not)/modify by working directly in the attached manuscript file.
- 2) We note that the quality of some images is not ideal, specifically Figures 2C and 3D,F. The resolution appears low and the images/lettering appear blocky/blurry even when observed at normal size. Please use better images as these issues could lead to problems when the production team tries to resize these images for the final manuscript.
- 3) The layout of panels D and F is a bit confusing (i.e. which blot belongs to which panel). Could you please reposition/rearrange? We also note the excessive contrasting applied to panel D.
- 4) We encourage the publication of source data, with the aim of making primary data more accessible and transparent to the reader. Would you be willing to provide a PDF file per figure that contains the original, uncropped and unprocessed scans of all or at least the key gels used in the manuscript and/or source data sets for relevant graphs? The files should be labeled with the appropriate figure/panel number, and in the case of gels, should have molecular weight markers; further annotation may be useful but is not essential. The files will be published online with the article as supplementary "Source Data" files. If you have any questions regarding this just contact me.

I look forward to reading a new revised version of your manuscript as soon as possible and in any case, within 2 weeks.

***** Reviewer's comments *****

Referee #2 (Remarks):

The authors did a nice job with this revision, which includes new data and insightful modification of the text.
Careful editing might however be necessary as some awkward sentences remain such as in lines 246-248.

Referee #3 (Remarks):

The authors have done a good job in addressing the points raised in my review. I recommend the revised paper for publication.

2nd Revision - authors' response

11 May 2017

Referee #2 (Remarks):

The authors did a nice job with this revision, which includes new data and insightful modification of the text.
Careful editing might however be necessary as some awkward sentences remain such as in lines 246-248.

Thank you for your suggestion. We checked the main text again to make it readable.

Referee #3 (Remarks):

The authors have done a good job in addressing the points raised in my review. I recommend the revised paper for publication.

Thank a lot.

Corresponding Author Name: Arne Holmgren, Jun Lu

Journal Submitted to: EMBO Molecular medicine

Manuscript Number: EMM-2017-07661